# Conditional Sampling of High-Quality Ultrasound Images from Diffusion Models with Limited Data

**Gurunath Reddy M**                    GURUNATHREDDY.M@GEHEALTHCARE.COM
**Sushanth Govinahallisathyanarayana** SUSHANTH.GOVINAHALLISATHYANARAYANA@GEHEALTHCARE.COM
**Prasad Sudhakar**                     PRASAD.SUDHAKAR@GEHEALTHCARE.COM
**Chandan Aladahalli**                  CHANDAN.ALADAHALLI@GEHEALTHCARE.COM
**Dattesh Shanbhag**                    DATTESH.SHANBHAG@GEHEALTHCARE.COM
*Advanced Technology Group, GE HealthCare, Bangalore, India*

## Abstract

Diffusion models can synthesize realistic medical images but are computationally intensive, requiring thousands of sampling steps. Additionally, since the conventional sampling process is stochastic, it can produce low-quality or irrelevant outputs, especially when trained with less data. In this work, we introduce a quality-aware sampling strategy that monitors image fidelity during generation, terminating trajectories that are likely to lead to low quality images early, and halting the process once satisfactory quality is achieved. This approach accelerates diffusion-based synthesis, decreases computational overhead, and yields anatomically plausible images that support the development of more robust healthcare AI models.

**Keywords:** Image quality, Diffusion models, Ultrasound, Synthetic data.

## 1. Introduction

AI models in healthcare rely on large, representative datasets. However, privacy restrictions, acquisition costs, and regulatory constraints make comprehensive data collection difficult. As a result, training sets often lack sufficient diversity, causing models to generalize poorly to even minor distribution shifts. Traditional augmentation methods—such as global changes in brightness, contrast, or sharpness—offer limited gains in robustness and fail to capture the fine-grained anatomical variations crucial for clinical accuracy. Generative diffusion models (Ho et al., 2020, 2022; Song and Ermon, 2019; Rombach et al., 2022; Gao et al., 2023) address these gaps by producing diverse, anatomically realistic medical images that can strengthen downstream tasks like classification and segmentation. However, their adoption is hindered by high computational demands: generating a single sample typically requires thousands of denoising steps. Additionally, the stochastic nature of the sampling process can yield suboptimal or irrelevant images even when conditioned with anatomical annotations such as segmentation masks. Mitigating the generation of low-quality outputs through repeated sampling further increases computational cost. In this work, we present a method for automatically generating high-quality images by enabling identification of high-fidelity synthetic images during sampling. Our approach reduces the number of sampling steps required and consequently reduces computational overhead. This results in faster, more efficient generation of diverse, anatomically plausible medical images. The proposed approach is placed in context of other approaches for accelerating diffusion models and improving sample quality in the related work section in the Appendix (see section 6.2).

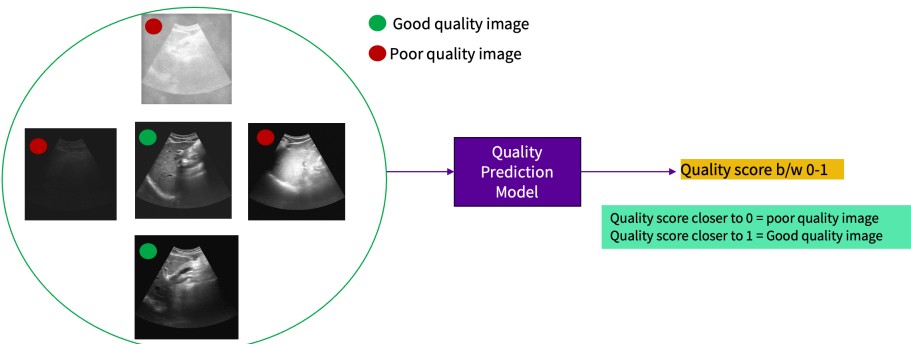

Figure 1: Quality prediction model trained on both high- and low-quality ultrasound images to estimate a quality score for each synthesized image.

## 2. Proposed Quality Aware Sampling

Diffusion model training involves forward and reverse diffusion process. The forward process progressively adds Gaussian noise to the original image, producing a sequence that approaches an isotropic Gaussian distribution while the reverse process aims to reconstruct the original image by iteratively denoising. The reverse process is controlled by incorporating information to condition the output distribution on anatomical constraints using segmentation masks. In this work, we train a conditional denoising diffusion model on abdominal ultrasound images primarily depicting the liver and portal vein. Since abdominal ultrasound information is confined within a sector, the model is additionally conditioned using a sector mask to ensure synthesis occurs only within this region.

### 2.1. Unguided Sampling-Initial Experiments

As a first step, we perform unguided sampling from the trained conditional diffusion model without any quality control or early stopping mechanism to assess the baseline quality of generated images. The conditional diffusion framework is based on a U-Net architecture and trained on 18,000 ultrasound images containing liver, portal vein, and sector masks for 200 epochs with 2,000 diffusion steps. During the synthesis of 3,406 images using the trained model, only 1,311 were of acceptable quality, while 2,095 exhibited poor fidelity (e.g., washed-out structures) as determined by the quality criteria (see appendix 5.2). This corresponds to approximately 61.5% unusable outputs. Given that sampling each image required about two minutes of GPU computation on A6000 GPU, this resulted in an estimated 69.83 GPU hours wasted on generating low-quality data.

### 2.2. Quality Prediction Model

Without a quality prediction model, the conditional diffusion model generates a mix of high and low-quality ultrasound images. We separate these outputs by computing image-quality metrics for each synthesized image: Laplacian variance (sharpness), global intensity standard deviation (contrast), and a noise estimate derived from the difference between the

image and its Gaussian-blurred counterpart (see Appendix 5.2). These metrics partition the synthesized images into good and poor-quality sets, which was further refined through manual inspection. Images deemed high quality are labeled as $y = 1$, while poor-quality images are labeled as $y = 0$. Using these labeled samples, we train a ResNet-18-based quality prediction model (Fig. 1) on a balanced dataset of 200 images. The model outputs a continuous quality score $\hat{y} \in [0, 1]$ for each synthesized image, enabling real-time quality-aware sampling and acceptance during generation.

### 2.3. Quality Aware Sampling

To improve efficiency during sampling, we integrate the quality scoring model into the iterative denoising process of the diffusion framework as shown in Fig. 5 in Appendix section. At each reverse step, the quality prediction model evaluates the partially denoised image and assigns a score. If this score stagnates or fails to improve over a predefined window of steps, the sampling process is terminated early and the image is discarded. Conversely, if the quality score is above the threshold within the monitoring window, the process halts and the image is retained as a high-quality output. This adaptive termination mechanism reduces redundant computation by avoiding full-length sampling for low-quality candidates, while ensuring that only high-fidelity synthetic images are synthesized.

### 3. Results and Discussion

Fig. 6 in Appendix section illustrates images generated by the diffusion model with and without the integration of the image quality prediction module. Without the quality model, many generated images appear washed out or noisy, whereas the quality-aware diffusion model produces images with better contrast and overall quality. Fig. 7 in Appendix section shows the progression of quality scores during the sampling process of the diffusion model integrated with the quality prediction module. In the top row, the quality model assigns near-zero scores to synthesized images of poor quality across all sampling steps when the denoising process fails to produce a high-quality image. In contrast, the bottom row demonstrates that the quality prediction model assigns progressively higher scores as image quality improves at each reverse denoising step. The mean quality metrics computed over 3,000 synthesized images for each diffusion model, both with and without the integration of the quality prediction module, are reported in Table 1 in the Appendix. As shown, incorporating the quality prediction model consistently leads to a significant improvement in the quality scores of the synthesized images compared to those generated by diffusion models without quality prediction.

### 4. Summary

This paper proposes a quality-aware sampling framework for conditional diffusion models to efficiently generate high-fidelity medical images under limited data conditions. A ResNet-18-based quality predictor monitors image fidelity during generation, enabling early termination of low-quality paths and reducing computational overhead.

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

## 5. Appendix

### 5.1. Training Conditional Diffusion Model on Ultrasound Images

Diffusion model training involves forward and reverse diffusion processes as shown in Fig. 2. The forward process gradually adds Gaussian noise to the original image $X_0$ over $T$ timesteps, producing a sequence $\{X_t\}_{t=1}^{T}$ that approaches an isotropic Gaussian distribution:

$$q(x_t \mid x_{t-1}) = \mathcal{N}\big(x_t; \sqrt{1 - \beta_t}\, x_{t-1}, \beta_t \mathbf{I}\big)$$

where:

- $\beta_t$ is the variance schedule for timestep $t$.

- $x_0$ is the clean ultrasound image.

- $x_T$ is nearly pure noise after $T$ steps.

The marginal distribution after $t$ steps can be expressed as:

$$q(x_t \mid x_0) = \mathcal{N}\big(x_t; \sqrt{\bar{\alpha}_t}\, x_0, (1 - \bar{\alpha}_t)\mathbf{I}\big)$$

where:

$$\bar{\alpha}_t = \prod_{s=1}^{t}(1 - \beta_s).$$

The reverse process aims to reconstruct $X_0$ from $X_T$ by iteratively denoising. The reverse transition is parameterized by a neural network $\epsilon_\theta$ that predicts noise:

$$p_\theta(x_{t-1} \mid x_t) = \mathcal{N}\big(x_{t-1}; \mu_\theta(x_t, t), \sigma_t^2\mathbf{I}\big)$$

where:

$$\mu_\theta(x_t, t) = \frac{1}{\sqrt{\alpha_t}}\Big(x_t - \frac{\beta_t}{\sqrt{1 - \bar{\alpha}_t}}\, \epsilon_\theta(x_t, t)\Big)$$

For conditional diffusion, the reverse process incorporates conditioning information $c$ (e.g., a mask of the portal vein) (see representative schematic in Fig.2 in appendix section 5.2):

$$p_\theta(x_{t-1} \mid x_t, c) = \mathcal{N}\big(x_{t-1}; \mu_\theta(x_t, t, c), \sigma_t^2\mathbf{I}\big)$$

This ensures the generated ultrasound image respects the anatomical constraints (shape, size, and location) specified by the mask. The conditional denoising diffusion model in this work is trained on ultrasound images primarily depicting the liver and portal vein, as illustrated in Fig. 3. Additionally, since abdominal ultrasound information is confined within a sector-shaped region, the model is conditioned using a sector mask to ensure synthesis occurs only within this region.

### 5.2. Quality Prediction Model

As discussed in the previous section, the conditional diffusion model generates a mix of high- and low-quality ultrasound images. To systematically separate these outputs, we first compute classical image-quality metrics for each synthesized image $I$: Laplacian variance (sharpness), global intensity standard deviation (contrast), and a noise estimate derived from the difference between the image and its Gaussian-blurred counterpart. Thresholding these metrics provides an initial partition into good and poor-quality sets, which is further refined through manual inspection. Images deemed high quality are labeled as $y = 1$, while poor-quality images are labeled as $y = 0$. Using these labeled samples, we train a ResNet-18-based quality prediction model (Fig. 1) on a balanced dataset of 200 images. The model outputs a continuous quality score $\hat{y} \in [0, 1]$ for each synthesized image, enabling real-time quality-aware sampling and acceptance during generation.

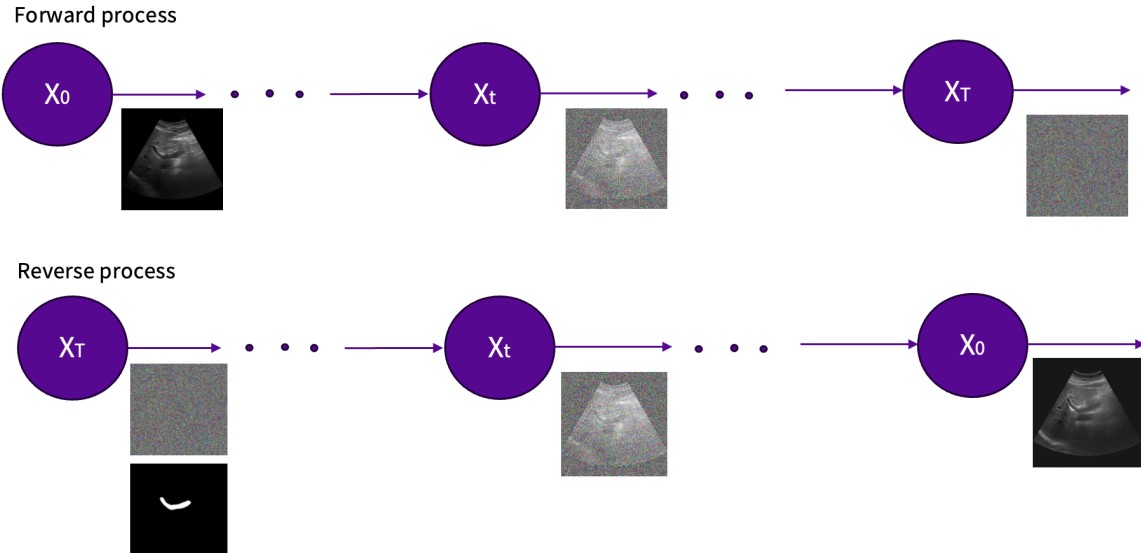

Figure 2: Illustration of the forward and reverse conditional diffusion process for ultrasound image synthesis. The forward process progressively adds noise to the original image, while the reverse process denoises step-by-step, conditioned on a portal vein mask to preserve anatomical structure.

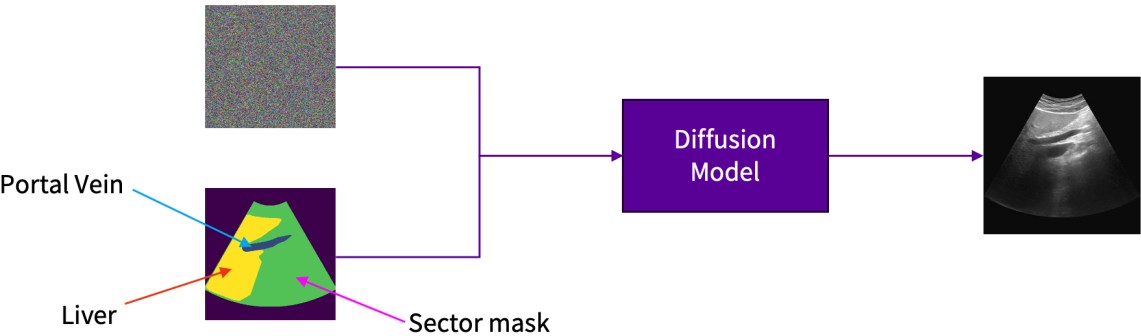

Figure 3: Diffusion model conditioned on portal vein, liver and sector mask to synthesize mask controlled ultrasound images.

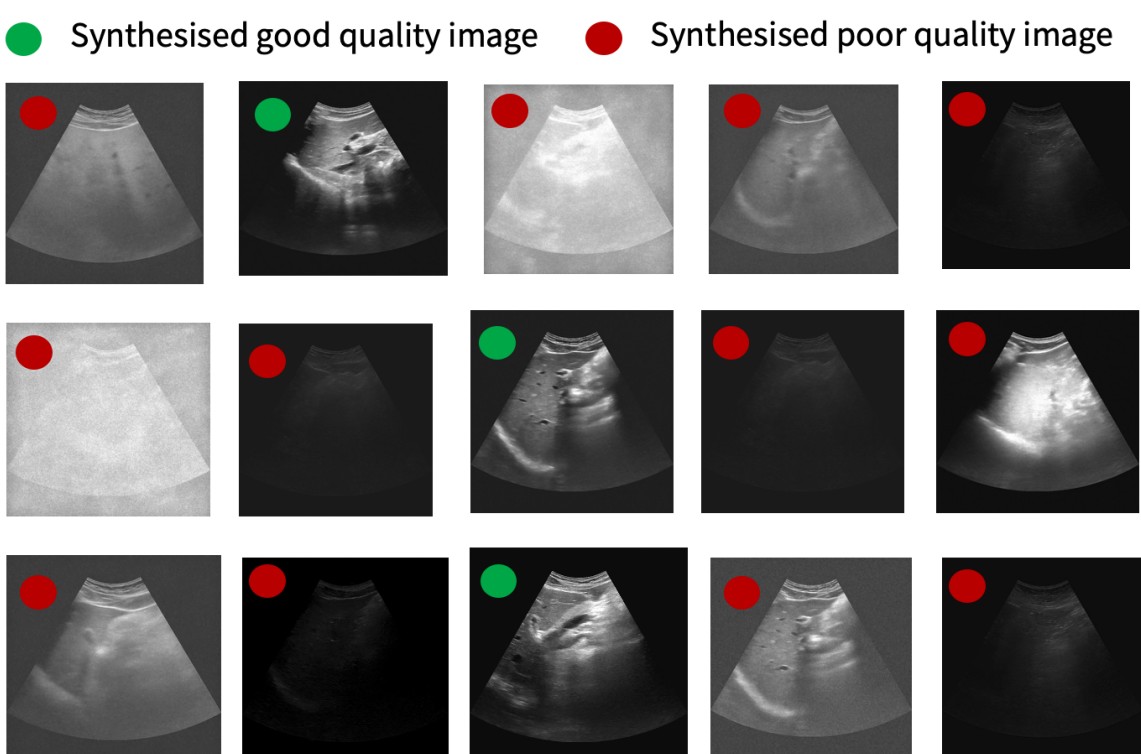

Figure 4: Diffusion model trained on limited data conditions generates mix of good and poor quality images.

**Image-Quality Metrics.** Let $I \in \mathbb{R}^N$ denote a synthesized image with $N$ pixels and mean intensity $\mu$. We define:

$$\text{Sharpness}(I) = \text{Var}(\nabla^2 I), \tag{1}$$

$$\text{Contrast}(I) = \sqrt{\frac{1}{N} \sum_{i=1}^{N} (I_i - \mu)^2}, \tag{2}$$

$$\text{Noise}(I) = \frac{1}{N} \sum_{i=1}^{N} \big| I_i - (G_\sigma * I)_i \big|, \tag{3}$$

where $\nabla^2 I$ is the Laplacian of $I$, $G_\sigma$ is a Gaussian kernel with standard deviation $\sigma$, and $*$ denotes convolution.

**Metric-Space Thresholding.** Given thresholds $\tau_S$, $\tau_C$, and $\tau_N$ for sharpness, contrast, and noise respectively, an image is classified as good quality if:

$$\text{Sharpness}(I) \geq \tau_S, \quad \text{Contrast}(I) \geq \tau_C, \quad \text{Noise}(I) \leq \tau_N. \tag{4}$$

The quality metric thresholds $\tau_S$, $\tau_C$, and $\tau_N$ were empirically determined based on an evaluation of visually good-quality images and set to 200, 20, and 0.3, respectively. An ultrasound image is classified as high quality if $\tau_N < 0.3$, $\tau_C > 20$, and $\tau_S < 200$.

**Quality Prediction Model.** Let $\mathcal{D} = \{(I_j, y_j)\}_{j=1}^{M}$ be the labeled dataset, with $y_j \in \{0, 1\}$ and $M = 200$. We train a ResNet-18 predictor $f_\theta : \mathbb{R}^N \to [0, 1]$ to output a quality score:

$$\hat{y}_j = f_\theta(I_j), \qquad \hat{y}_j \in [0, 1]. \tag{5}$$

The model is optimized using binary cross-entropy:

$$\mathcal{L}_{\text{BCE}}(\theta) = -\frac{1}{M} \sum_{j=1}^{M} \Big[ y_j \log \hat{y}_j + (1 - y_j) \log(1 - \hat{y}_j) \Big]. \tag{6}$$

**Quality-Aware Acceptance.** During sampling from the proposed quality aware diffusion model, the predicted score $\hat{y}$ is used to accept or discard samples in real time:

$$\text{Accept}(I) \iff \hat{y} \geq \tau_q, \tag{7}$$

where $\tau_q \in (0, 1)$ is a user-defined threshold. The threshold $\tau_q$ is set to 0.95 to ensure that only high-quality images are synthesized with high confidence. This strategy minimizes redundant computation and ensures that only high-fidelity synthetic images are retained.

### 5.3. Integrating Quality Prediction Model into Diffusion Model

To improve efficiency during sampling, we integrate the quality scoring model into the iterative denoising process of the diffusion framework as shown in Fig. 5. At each reverse step $t$, where the model predicts $\epsilon_\theta(x_t, t)$ and updates the mean $\mu_\theta(x_t, t)$ as:

$$p_\theta(x_{t-1} \mid x_t) = \mathcal{N}\big(x_{t-1}; \mu_\theta(x_t, t), \sigma_t^2 \mathbf{I}\big)$$

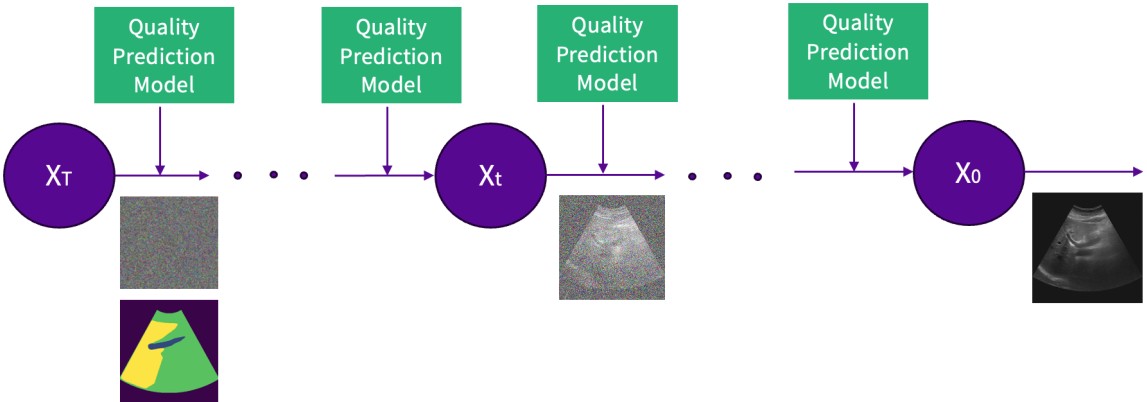

Figure 5: Integration of quality prediction module within the diffusion framework to enable real-time synthesis of high-fidelity ultrasound images.

$$\mu_\theta(x_t, t) = \frac{1}{\sqrt{\alpha_t}}\left(x_t - \frac{\beta_t}{\sqrt{1-\bar{\alpha}_t}}\,\epsilon_\theta(x_t, t)\right)$$

the quality prediction model evaluates the partially denoised image $x_t$ and assigns a score $\hat{y}_t \in [0, 1]$. If this score stagnates or fails to improve over a predefined window of steps, the sampling process is terminated early and the image is discarded. Conversely, if $\hat{y}_t \geq \tau_q$ (a user-defined threshold) within the monitoring window, the process halts immediately and the image is retained as a high-quality output:

$$\text{Accept}(x_t) \;\Leftrightarrow\; \hat{y}_t \geq \tau_q.$$

This adaptive termination mechanism reduces redundant computation by avoiding full-length sampling for low-quality candidates, while ensuring that only high-fidelity synthetic images are preserved for downstream tasks.

Table 1: Image Quality Metrics Comparison

|  | Noise↓ | Contrast↑ | Laplacian Variance↓ |
|---|---|---|---|
| Without Quality Prediction | 0.31 | 28.82 | 469.87 |
| With Quality Prediction | 0.26 | 30.34 | 147.34 |

## 6. Related Work and Contributions

Diffusion models trained under limited data conditions often generate a mix of high- and low-quality images as shown in Fig. 4. In large-scale synthetic data generation scenarios, inefficiencies in producing low-quality samples can significantly increase computational and operational costs. Our proposed quality-aware sampling strategy mitigates this issue by dynamically identifying high-fidelity images during generation, thereby reducing redundant computation and ensuring that only high-quality synthetic data is retained for training downstream models. Others have explored various strategies to accelerate diffusion models

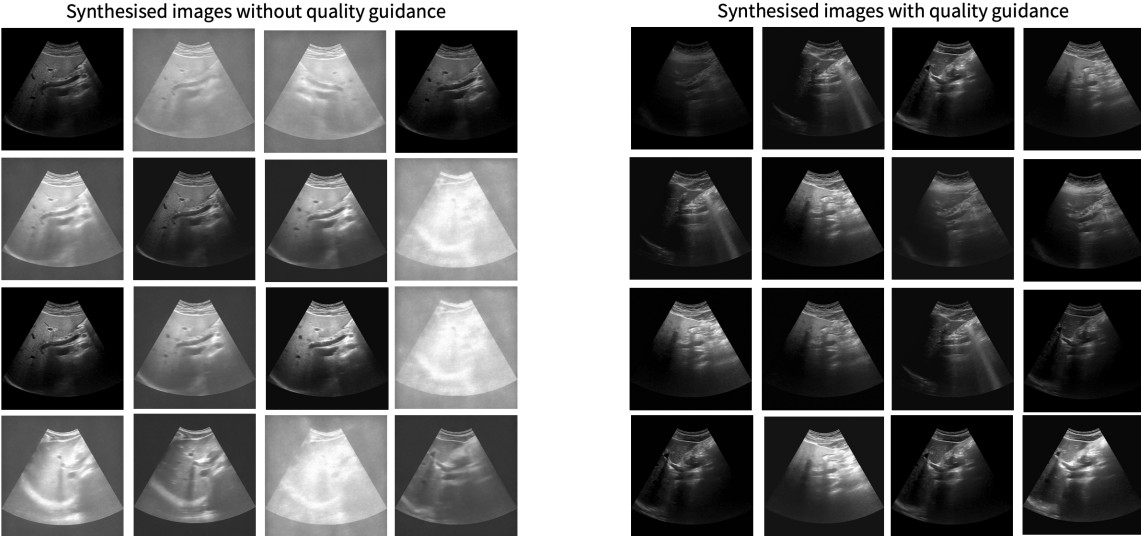

Figure 6: Synthesized images from a diffusion model with and without the integration of the quality prediction module.

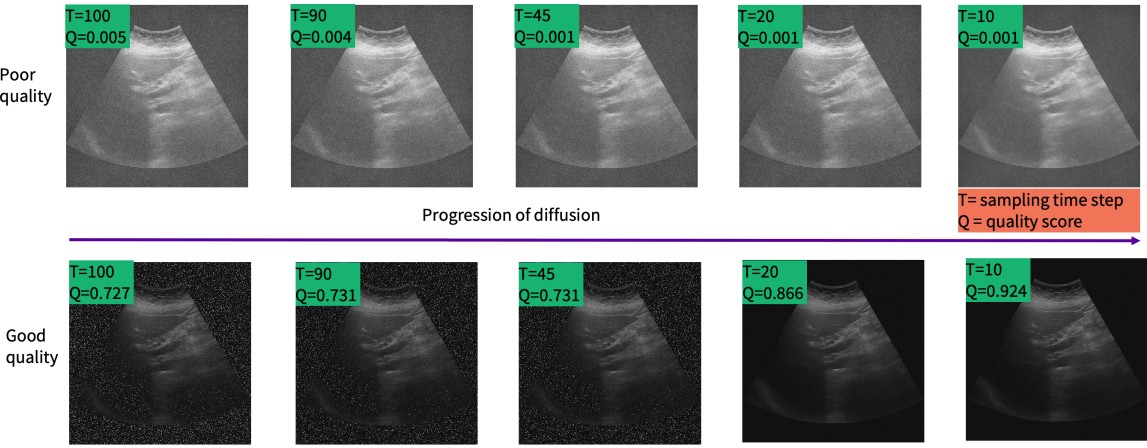

Figure 7: Progression of quality scores across diffusion sampling steps.

and improve sample quality as outlined below. However, our approach is distinct in its real-time monitoring of image fidelity during the sampling process. Comparison of our method with existing approaches is summarized as follows:

1. Real-Time Quality Monitoring vs. Fixed Sampling Strategies: Our approach uses a quality scoring model to evaluate image fidelity during generation. Sampling is dynamically adjusted based on the evolution of the image quality score. Existing methods, such as Early-Stopped DDPM (ES-DDPM) (Lyu et al., 2022), reduce sampling steps by starting the reverse process from a non-Gaussian distribution, often using pre-trained GANs or VAEs. However, they do not monitor image quality during generation; the stopping point of generating is predefined and static.

2. Adaptive Computation vs. Quality-Based Termination: AdaDiff (Dhariwal and Nichol, 2021) introduces step-wise adaptive computation using a timestep-aware uncertainty estimation module (UEM). It adjusts computational effort per step but focuses on uncertainty, not image quality. Our method is quality-centric, using a scoring model to actively decide whether to continue or stop generation, which is more aligned with the goal of producing high-fidelity outputs efficiently.

3. Application to Medical Imaging: Diffusion models have previously been used to generate synthetic medical images for training CNNs (Moon et al., 2023; Ma et al., 2024). However, these works do not optimize the sampling process based on image quality during generation. Our method directly addresses quality assurance during generation, which is critical for medical applications where fidelity and diversity are paramount. Additionally, our method is able to provide this improved fidelity to the data distribution at a reduced computational burden.

4. Integrated Quality and Semantic Evaluation: SF-IQA (Chen et al., 2025) introduces a metric that combines image quality and text-image similarity for evaluating AI-generated images. However, it is post-hoc—used after generation for assessment. Our solution integrates quality scoring within the generation loop, enabling real-time decision-making and resource optimization.

In summary, our key contributions are: Integration of Quality Scoring: Embedding a quality scoring model directly into the diffusion sampling loop. Real-Time Fidelity-Based Control: Enabling dynamic early termination or acceptance decisions based on real-time image quality assessment. Efficiency Gains: Reducing computational overhead while preserving high image fidelity. Domain-Specific Impact: Tailoring the approach for medical imaging, where generating high-quality synthetic data is critical for downstream tasks.

