# OpenReview forum: "Conditional Sampling of High-Quality Ultrasound Images from Diffusion Models with Limited Data"
_MIDL.io/2026/Short_Papers — MIDL 2026 - Short Papers Poster_

### Official Review · Reviewer_GQiK · 2026-04-29
**Image quality guidance for diffusion models**

**Rating:** 2
**Confidence:** 5

**Review:**

Please see Strengths and Weaknesses below.

**Summary:**

The authors introduce quality-guided sampling for diffusion models to encourage the models to deviate from low-quality sampling trajectories. The guidance is completed via an auxiliary classifier trained to differentiate between images according to noise, contrast, etc.-based statistics. The authors test on an ultrasound dataset, and find improved sampling quality according to similar statisics/metrics.

**Strengths:**

Training strong generative medical image models on small datasets is an ever-present challenge, so the main task is important. The core idea of this proposal, guiding medical image generation more carefully to avoid low-quality pitfalls, is reasonable.

**Weaknesses:**

This submission has major weaknesses:
- The quality detector's labeled examples for training are based on relatively simple image features/metrics, so it is unsurprising that incorporating the detector results in generated images improving on these exact same metrics. Therefore, the method requires better validation metrics to indicate that it would actually be useful in practice, namely downstream-task based metrics (e.g. does training some downstream model on images generated with vs. without quality guidance results in better performance?), or less importantly, typical perceptual generative model metrics such as FID, KID, RadiologyFID, FRD (see e.g. Konz and Osuala et al, Medical Image Analysis 2026 for a recent survey). Without these results, its not yet clear to me that this is a useful contribution. While only testing on one dataset/modality is OK for MIDL short papers, these metrics are crucial.
- The paper is quite reliant on the appendix; the paper should stand on its own without the appendix (the appendix should just be supplementary), so the main method details and results need to be presented in the main paper. Similar important details should also be in the main text, e.g. details of the dataset (which I don't see anywhere in the main text or supplementary). Without the appendix, many key method, dataset, result, etc. details are not explicitly present whatsoever.
- In the introduction, "However, their adoption is hindered by high computational demands: generating a single sample typically requires thousands of denoising step". This limitation isn't really strong any more due to fast generation models e.g. DDIM, flow-matching models, consistency models, very recently drift models, etc, which also might avoid this "bad trajectory" problem, so it hurts the relevance of this work. I encourage the authors to discuss this and perform a recent literature review.

**Justification Of Rating:**

Because of the major weaknesses I discussed, I'm not sure what a clear, relevant-to-the-state-of-the-art, useful contribution is in this paper. MIDL short papers only need to have preliminary results, but the results still should show promising usefulness, impact, and interest to the MIDL community.

---

### Decision · Program_Chairs · 2026-05-08

Accept (Poster)